Single-cell transcriptome analysis reveals cellular reprogramming and changes of immune cell subsets following tetramethylpyrazine treatment in LPS-induced acute lung injury

Wu Mingyan
Wang Shanmei
Chen Xiaolan
Shen Li
Ding Jurong
Jiang Hongbin jianghongbin10@163.com
Department of Emergency Medicine, Shanghai Pulmonary Hospital, Tongji University School of Medicine , Shanghai , China
Zhang Xin
Electronic publication date: 2025 Jan 13
Publication date: 2025
Volume: 13
Electronic Location ID: e18772
Received 2024 Jun 10; Accepted 2024 Dec 5
Copyright: © 2025 Wu et al.
Copyright year: 2025
Copyright holder: Wu et al.
License: This is an open access article distributed under the terms of the Creative Commons Attribution License, which permits unrestricted use, distribution, reproduction and adaptation in any medium and for any purpose provided that it is properly attributed. For attribution, the original author(s), title, publication source (PeerJ) and either DOI or URL of the article must be cited.
License URL: https://creativecommons.org/licenses/by/4.0/

Keywords: Acute lung injury, Tetramethylpyrazine, scRNA-seq, Lipopolysaccharide, FOSL1, JunB

Funding: Shanghai Municipal Level Hospital Emergency Medicine Specialized Alliance Promotion and Improvement Planning SHDC22021304 This work was funded by Shanghai Municipal Level Hospital Emergency Medicine Specialized Alliance Promotion and Improvement Planning (SHDC22021304). The funders had no role in study design, data collection and analysis, decision to publish, or preparation of the manuscript.

==============================
Background

Acute lung injury (ALI) is a disordered pulmonary disease characterized by acute respiratory insufficiency with tachypnea, cyanosis refractory to oxygen and diffuse alveolar infiltrates. Despite increased research into ALI, current clinical treatments lack effectiveness. Tetramethylpyrazine (TMP) has shown potential in ALI treatment, and understanding its effects on the pulmonary microenvironment and its underlying mechanisms is imperative.

Methods

We established a mouse model of lipopolysaccharide (LPS)-induced ALI and performed single cell RNA sequencing (scRNA-seq). Bioinformatic analyses of the immune, epithelial and endothelial cells were then performed to explore the dynamic changes of the lung tissue microenvironment. We also analyzed the effects of TMP on the cell subtypes, differential gene expression and potential regulation of transcriptional factors involved. Immunohistochemistry and enzyme-linked immunosorbent assay were performed to identify the effects of TMP on immune inflammatory response.

Results

We found that TMP efficiently protected against LPS-induced acute lung injury. Results of scRNA-seq showed that the cells were divided into seven major cell clusters, including immune cells, fibroblasts, endothelial cells and epithelial cells. Neither dexamethasone (Dex) nor TMP treatment showed any significant protective effects in these clusters. However, TMP treatment in the LPS-induced ALI model significantly increased follicular helper T cells and reduced CD8+ naive T cells, Vcan-positive monocytes and Siva-positive NK cells. In addition, TMP treatment increased the number of basal epithelial cells and lymphatic endothelial cells (LECs), indicating its protective effects on these cell types. Scenic analysis suggested that TMP likely mitigates LPS-induced injury in epithelial and endothelial cells by promoting FOSL1 in basal epithelial cells and JunB in LECs.

Conclusions

Our findings suggest that TMP appears to alleviate LPS-induced lung injury by regulating the immune response, promoting epithelial cell survival and boosting the antioxidant potential of endothelial cells. This study highlights the potential therapeutic use of TMP in the management of ALI.

Introduction

Acute lung injury (ALI) is a severe pulmonary disorder characterized by diffuse alveolar damage, leading to excessive inflammation and cell death (Xiao et al., 2020; Favarin et al., 2013). Prolonged and repetitive injury triggers tissue and cellular responses that ultimately result in pulmonary fibrosis (Thannickal et al., 2014). Alterations in endothelial and epithelial cells are critical features of acute alveolar injury in ALI. Integrity of the epithelial/endothelial barrier could prevent the flooding of the alveolar space with high molecular weight proteins, alveolar edema formation and leukocyte extravasation (Schulte et al., 2011). The endothelial cells play important roles in limiting vascular hyperpermeability and leukocyte infiltration. The epithelial cell monolayer stabilizes the alveolar barrier, producing surfactant and ensuring alveolar fluid clearance (Long, Mallampalli & Horowitz, 2022; Peters et al., 2014). However, severe ALI is characterized by an increase in lung vascular permeability and the disruption of the alveolar-capillary barrier function due to either the death or dysfunction of alveolar epithelial cells and/or pulmonary capillary endothelial cells (Matthay et al., 2019).

Patients with ALI consistently fall into the so-called hyper-inflammatory sub-phenotype, which is characterized by high plasma levels of inflammatory biomarkers such as tumor necrosis factor-α (TNF-α), interleukin-6 (IL-6), interleukin-1β (IL-1β) (He et al., 2021; Li et al., 2023). It has been reported that various immune cells contribute to ALI pathogenesis. This includes myeloid cells that secrete pro-inflammatory mediators to induce host immune defense and M1 macrophages that can facilitate viral replication (Sefik et al., 2022; Lv et al., 2021). The blocking of proinflammatory functions in the Ly6C+ subgroup of CD8+ T cells by mesenchymal stem cells has been shown to alleviate lipopolysaccharide (LPS)-induced ALI (Zhu et al., 2020). Recent studies have identified several molecular mechanisms and potential therapeutic targets, such as the epithelial-mesenchymal transition (EMT) in the initiation and progression of lung fibrosis, YAP1’s role in alleviating sepsis-induced ALI by inhibiting ferroptosis and the effects of TNF-α inhibition of pulmonary capillary endothelial permeability (Thiery et al., 2009; Zhang et al., 2022; Fang et al., 2018). Despite these advances, the primary clinical approach for treating ALI remains symptomatic supportive care due to the lack of clinically approved targeted treatments, highlighting the need for further research to address the complex pathophysiology of ALI.

Ligustrazine, also known as 2,3,5,6-tetramethylpyrazine (TMP), is a major active ingredient derived from the Szechwan lovage rhizome of a Chinese plant (Gao et al., 2015). TMP possesses anti-inflammatory and antioxidative benefits (Chen et al., 2006; Sue et al., 2009) and exerts anti-apoptotic effects by suppressing caspase-3 activation pathways (Gao et al., 2015). Recent research has focused on the complex mechanisms underlying the protective effects of TMP. For instance, TMP has been shown to enhance the migration of bone marrow-derived mesenchymal stem cells and improve cerebral ischemia recovery in rats (Li et al., 2017). It has also been found to alleviate LPS-induced ALI by inhibiting alveolar macrophage pyroptosis and apoptosis (Jiang et al., 2021), as well as by blocking Rho/ROCK and Rac1/LIMK1 pathways in endothelial cells (Chen et al., 2019; Min et al., 2023). However, the dynamic changes in the cellular microenvironment during TMP treatment of ALI remain poorly understood, which limits the elucidation and treatment of ALI with TMP. Therefore, the present study was undertaken to investigate the dynamic variations in the cellular microenvironment during the treatment of ALI with TMP.

In the present study, we aimed to determine the multifaceted protective effects of TMP against lung injury. Our results showed that TMP suppressed the activation of CD8+ naive T cells and increased the number of follicular helper T cells in an LPS-induced mouse model of ALI. Additionally, TMP reduced the number of Vcan-positive monocytes and Siva-positive NK cells. Furthermore, TMP increased the numbers of basal epithelial cells and lymphatic endothelial cells (LECs) by inducing the expression and activation of FOSL1 and JunB, respectively. These findings indicate that TMP plays important roles in regulating immune inflammatory response and cell survival against injury in different cell types, suggesting it has potential therapeutic use in the management of ALI. This study provides a fundamental understanding of the possible molecular mechanisms underlying the effects of TMP and its potential clinical applications in the treatment of patients with ALI.

Materials and Methods

The LPS-induced ALI model in mice

A total of 12 six-to-eight-week adult male C57BL/6J mice (18–22 g) were purchased from the Shanghai Model Organisms Center, Inc. and housed in a specific-pathogen-free facility under a light/dark cycle of 12 h/12 h, with access to food and water ad libitum. The mice were housed at a density of five mice per cage to ensure adequate space for activity and optimal housing conditions. No mice were excluded from the analysis. The experimental procedures were conducted in accordance with the guidelines and the Animal Welfare Act Regulations of Shanghai Pulmonary Hospital, Tongji University. The animal treatment protocol followed the methods described in a previous study (Jiang et al., 2021). Briefly, the mice were randomly divided into four groups of three mice: a control group (injected intraperitoneally with 0.2 ml of 0.9% saline) and a model group administered with intraperitoneal injection of LPS in saline (0.2 ml) at a dosage of 30 mg/kg to induce ALI. The LPS+Dex group and LPS+TMP group were treated with intraperitoneal injection of either 0.2 ml of Dex (5 mg/kg) or 0.2 ml of TMP (50 mg/kg) 1 h before LPS administration. After a 72-h treatment period, the mice were anesthetized with an intraperitoneal injection of ketamine (80 mg/kg) and xylazine (10 mg/kg), then lung tissues were collected for subsequent analysis. All the animals were bred under specific pathogen-free conditions Shanghai Pulmonary Hospital Laboratory Animal Center, in accordance with hospital guidelines (Ethics approval number: Q20-269Y). All animal studies were approved by the Institutional Animal Care and Use Committee of Tongji University.

Histological analysis

Lung tissues were isolated, fixed in 4% paraformaldehyde for 48 h and dehydrated using different concentrations of ethanol solution, from 70% to 100% for 40 min. The samples were then embedded in paraffin and cut into 4 μm sections. Hematoxylin-eosin (H & E) staining was performed on the sections. The slides were dyed with hematoxylin for 5 min and then they were rinsed with water, and then after returning to a blue coloration, the slides were stained with eosin for 1–3 min. The degree of lung damage was assessed by two independent technicians who were blinded to the experimental groups using a recently published criterion (Qiang et al., 2020). The histological images were observed using an optical microscope (Nikon, Tokyo, Japan).

Enzyme-linked immunosorbent assay

Blood samples were obtained from the orbital sinus and these were allowed to clot for 30 min at 37 °C. The samples were then centrifuged at 3,500 rpm for 10 min at 4 °C. The concentrations of IL-1β and TNF-α were determined using commercially available enzyme-linked immunosorbent assay (ELISA) kits (Abcam, Cambridge, United Kingdom), by following the manufacturer’s protocols. The absorbance at 450 nm was measured at the end of the procedures using a microplate reader.

Cell counting in BALF

The mice were anesthetized, and a surgical incision was made in the anterior neck to expose the trachea. Bronchoalveolar lavage fluid (BALF) was collected by instilling and withdrawing 0.5 mL of saline three times through an endotracheal tube. The collected BALF samples were combined and centrifuged at 3,500 rpm for 10 min at 4 °C. The resulting cell pellet was resuspended, and red blood cells were removed by using a lysis buffer. This process was repeated twice, with centrifugation at 3,500 rpm for 10 min at 4 °C each time. The final cell pellet was resuspended in 100 µL of saline. The cell types were identified and then they were quantified by using Wright-Giemsa staining.

Immunohistochemistry

The lung tissues were fixed in formalin, embedded in paraffin and then they were sectioned to 4 µm thickness. The sections were mounted on glass slides and heated at 65 °C overnight. After de-paraffinization and rehydration, the endogenous peroxidase activity was blocked by using 3% hydrogen peroxide for 20 min. To minimize non-specific binding, the slides were incubated with 5% normal goat serum for 1 h at room temperature. The sections were then incubated overnight at 4 °C with primary antibodies against CD45, CD8 and IL-6 (ab40763, ab217344 and ab9324, respectively; Abcam, Cambridge, UK) at a 1:250 dilution. After washing with TBST, the slides were incubated with HRP-conjugated goat anti-rabbit secondary antibody for 1 h at room temperature. The sections were then stained with DAB and counterstained with hematoxylin.

Single-cell capture, cDNA library preparation and sequencing processing

Twelve mice were randomly assigned to four experimental groups: Control, LPS, LPS+Dex, and LPS+TMP for single-cell transcriptomic analysis. Single-cell sequencing achieved an average depth of 31,987 reads per cell. The statistical power of this experimental design, calculated using RNASeqPower (https://rodrigo-arcoverde.shinyapps.io/rnaseq_power_calc/), was determined to be 0.91. Fresh lung tissues were resected from mice and stored in tissue preservation solution (21903-10; Shanghai Biotechnology Corporation, Shanghai, China). The tissues were placed into a sterile culture dish and rinsed with 5 mL of 1 X DPBS precooled at 4 °C. This was repeated 1–2 times to remove any residual tissue protective solution. The moistened tissues were placed in a new 1.5 mL EP tube with 500 μL of digestive solution and then these were cut into small pieces. The shredded tissues were transferred into a 50 mL centrifuge tube containing 5 mL of digestive solution, and this was gently shaken at 37 °C in a water bath. The first sample was collected after 15 min of digestion, and fresh digestive solution was added and the tissues was digested for a further 20 min. The digested cell suspensions were pooled and these were passed twice through pre-wetted 40 μm filter membranes. The filtrates were collected into new 50 mL centrifuge tubes and centrifuged at 2,000 rpm at room temperature for 10 min. The supernatants were discarded and 2 mL of precooled red blood cell lysis solution was added to the pellets. The cell suspensions obtained were gently dispersed and they were incubated at room temperature for 3 min. After the red blood cells were lysed, the remaining cells were centrifuged at 1,500 rpm for 5 min at room temperature. The supernatants were discarded and the cell pellets were re-suspended in PBS containing 2% FBS.

The dissociated cells were then stained with 0.4% trypan blue (Cat. no. 14190144; Thermo Fisher Scientific, Waltham, MA, USA) to check for viability on a Countess® II Automated Cell Counter (Thermo Fisher Scientific, Waltham, MA, USA) and then the number of cells obtained were counted. A total of 10× library preparation and sequencing bead solution with unique molecular identifier (UMI) and cell barcodes was loaded into the cell suspension close to the saturation point. This ensured that each cell was paired with a bead in a Gel Beads-in emulsion (GEM). After addition of cell lysis buffer, the poly-adenylated RNA molecules hybridized to the beads. The beads were retrieved into a single tube for reverse transcription. After cDNA synthesis, each cDNA molecule was tagged at the 5′-end (that is, the 3′-end of a messenger RNA transcript) with an UMI and a cell label indicating its cell of origin. The 10× beads that were obtained were then subjected to second-strand cDNA synthesis, adaptor ligation and universal amplification. Sequencing libraries were prepared using randomly interrupted whole-transcriptome amplification products to enrich the 3′-ends of the transcripts linked with the cell barcode and UMI. All the remaining procedures, including the library construction, were performed according to the standard manufacturer’s protocol (CG000206 RevD). The sequencing libraries were quantified using a High Sensitivity DNA Chip (Agilent, Santa Clara, CA, USA) on a Bioanalyzer 2100 and the Qubit High Sensitivity DNA Assay (Thermo Fisher Scientific, Waltham, MA, USA). The libraries were sequenced on a NovaSeq 6000 (Illumina, San Diego, CA, USA) system by using 2x150 chemistry.

Single-cell RNA-seq data processing and filtering

The single-cell (sc) RNA-seq data were processed using the Cell Ranger 3.0.1 pipeline. FASTQs generated from the Illumina sequencing output were aligned to the mouse genome, version mm10, using the STAR algorithm. Next, the gene-barcode matrices for each sample was generated by counting UMIs and filtering the non-cell associated barcodes. Finally, we generate a gene-barcode matrix containing the barcoded cells and gene expression counts. This output was then imported into the Seurat (v3.0.2) R toolkit for quality control and downstream analysis of our scRNA-seq data. All functions were run with default parameters, unless otherwise specified. We excluded cells with fewer than 200 or more than 5,000 detected genes, and selected each gene had to have at least one UMI aligned in at least three cells. The expression of mitochondria genes was calculated using the PercentageFeatureSet function in the Seurat software package. To remove low activity cells, those with more than 10 percent of mitochondrial expressed genes were excluded. The normalized data (obtained using the NormalizeData function in the Seurat package) was performed to extract subsets of variable genes. The variable genes were identified by controlling for the strong relationship between variability and average expression. Next, we integrated the data from different samples after identifying ‘anchors’ between our datasets by using the FindIntegrationAnchors and IntegrateData in the Seurat software package.

Clustering, visualization and identification of cell types

We performed principal component analysis (PCA) and reduced the data to the top 30 PCA components after scaling the data. We visualized the clusters on a 2D map produced by using uniform manifold approximation and projection (UMAP). Identification of the cell types and subtypes was by nonlinear dimensional reduction. Cells were clustered using graph-based clustering of the PCA reduced data by using the Louvain method after computing a shared nearest neighbor graph. For sub-clustering, we applied the same procedure of scaling, dimensionality reduction and clustering to the specific set of data and this process was usually restricted to one type of cell). For each cluster, we used the Wilcoxon rank-sum test to find the significant deferentially expressed genes when comparing the remaining clusters. SingleR and known marker genes were used to identify the cell types in our samples.

Statistical analysis

Statistical analysis was conducted using the R 4.0.3 software package. The two-sided Wilcoxon rank-sum test and the one-way ANOVA followed by Dunnett’s test were used to evaluate the statistical significance between groups. A p-value of 0.05 or less was considered statistically significant.

Results

LPS-induced ALI is mitigated by treatment with TMP

We first established a model of ALI by utilizing LPS and administered either Dex or TMP to determine their protective effects. The traditional anti-inflammatory drug Dex was used as a positive control. Subsequently, we collected lungs and dissociated tissues for scRNA-seq analysis. Integrated analysis and expression profiling was conducted to investigate the potential mechanism of TMP protection against ALI. Histological examination with H & E staining demonstrated that LPS-induced ALI, characterized by pulmonary alveolar wall thickening and inflammatory cell infiltration, was attenuated significantly by either Dex or TMP treatment (Fig. 1A). We also determined the protective effects of TMP on the immune inflammatory response in the lung tissues following LPS treatment, by staining for immune cell markers and measurement of cytokine production. The results showed that the LPS induced the expression of CD45, CD8 and IL-6, which was significantly mitigated by treatment with either Dex or TMP (Fig. 1B). Consistent results were obtained when determining the levels of inflammation-related cytokines in the mouse blood samples. We found that the concentrations of TNF-α and IL-1β in the blood, which were significantly raised by LPS, were subsequently decreased by treatment with TMP (Figs. 1C and 1D). In addition, LPS led to a significant increase of total cell number in the BALF, which was attenuated by treatment with either Dex or TMP (Fig. 1E). These results indicated that TMP could efficiently protects against LPS-induced ALI.

Figure 1 Treatment with TMP attenuated LPS-Induced acute lung injury.

(A) Hematoxylin and Eosin histology results showing that LPS-induced acute lung injury was mitigated by Dex or TMP treatment. Scale bar = 100 μm for the upper image and Scale bar = 25 μm for the lower image. (B) LPS-induced the expression of CD45, CD8 and IL-6 was significantly decreased by treatment with either Dex or TMP. Scale bar = 100 μm. (C and D) The concentrations of TNF-α (C) and IL-1β (D) in blood samples, which were significantly raised by LPS, were decreased by treatment with TMP. (E) LPS lead to significant increases in total cell number in BALF, which was attenuated by treatment with either Dex or TMP. An asterisk (*) indicates p < 0.05.

scRNA-seq analysis of LPS-damaged lung tissues after TMP treatment

After removing low-quality scRNA-seq cells and reducing the possible batch effects by using the Harmony algorithm, we obtained 116,213 high-quality scRNA-seq profiles of various cell types from all our samples. As shown in Fig. 2A, uniform manifold approximation and projection (UMAP) analysis of cell types revealed seven major cell clusters: myeloid cells (48,650), B cells (30,458), T cells (12,072), NK cells (10,605), fibroblasts (6,252), endothelial cells (4,101) and epithelial cells (4,065) (Fig. 2A). The canonical marker genes used for dividing the major cell clusters are presented in Figs. 2B and 2C. We found that there were no unique cell subpopulations and the seven major cell clusters were present in different groups (Figs. 2D and 3A). Quantification of the cell clusters revealed that LPS significantly increased the percentage of myeloid and NK cells, while reducing the percentage of B and T cells. However, either Dex or TMP treatment did not show any protective effects against LPS with respect to the cell cluster percentages. Similar results were observed for the percentage of endothelial and epithelial cells (Figs. 3B and 3C). These results suggest that TMP may exert its protective effects by influencing specific cell subpopulations.

Figure 2 scRNA-seq analysis in lung tissues of all our samples.

(A) Uniform manifold approximation and projection (UMAP) embedding of single-cell profiles of lung tissues. Seven major cell clusters were distinguished and they are represented by different colors. (B and C) The major cell type marker genes for each cell clusters were represented (B) and expressed (C) in the UMAP. (D) The absence of unique cell subpopulations among different groups.

Figure 3 Quantification of the diverse cell types among different groups.

(A) The presence of the seven major cell clusters in all four subject groups. (B and C) Quantification of the proportions of cell clusters (B), and no significant protective effects of either Dex or TMP against LPS in the seven major cell clusters were observed (C).

Identification of changes in immune cell subsets after TMP treatment in LPS-induced ALI

To investigate whether any immune cell sub-clusters were involved in TMP protection against the immune response induced by LPS, we classified the three immune cell types (T, myeloid and NK cells) into sub-clusters (Ren et al., 2021). As shown in Fig. 4A, the T cells were divided into nine sub-clusters based on the marker genes, and a heatmap analysis of the top three genes for each T cell cluster is presented in Figs. 4B. We found that LPS significantly increased the percentage of CD8+ naive T (CD8+Tn) cells, while decreasing the percentage of follicular helper T (Tfh) cells. Treatment with either Dex or TMP significantly decreased the number of CD8+Tn cells, with both drugs having similar inhibitory effects. TMP treatment also increased the number of Tfh cells when compared to the LPS group, while Dex did not have this effect (Fig. 4C). In other cell sub-clusters, LPS also increased the cell number of Double Negative T cells (DNT), Effector T cells (Teff) and Regulatory T cells (Treg), whereas there were no significantly inhibitory effects of TMP on LPS-induced cell number change found in these cell subsets (Fig. S1).

Figure 4 Differential remodeling of T cells in LPS-induced acute lung injury after TMP treatment.

(A and B) Sub-clustering of T cells based on marker genes (A). Heatmap showing the top three genes for each T cell sub-cluster (B). (C) The LPS-induced increase in CD8+ naive T (CD8+Tn) cells and decrease in follicular helper T (Tfh) cells, which were reversed by TMP treatment.

Additionally, the myeloid cells were divided into 11 sub-clusters (Fig. 5A). The top three genes for each cell clusters of myeloid cells are presented in a heatmap (Fig. 5B). We found that LPS-increased a subset of cells within the C3 sub-cluster, named the Vcan+ monocyte, and this was reduced by Dex treatment. However, TMP had an only slight inhibitory effect on the percentage of the Vcan+ monocytes (Fig. 5C). In the other cell subsets of myeloid cells, LPS led to increased percentages of inflammatory Macrophages (Inflam-Mφ) and resident Tissue Macrophages (RTM-Mφ) cells, whereas no significantly inhibitory effects of TMP were found in these two cell subsets (Fig. S2). Furthermore, the NK cells were divided into 10 sub-clusters (Fig. 6A), and heatmap analysis of the top three genes for each of these was performed (Fig. 6B). LPS significantly induced the Siva+ NK cells and this is a subset of cells within the C5 sub-cluster. This effect was attenuated by either Dex or TMP treatment, particularly by Dex treatment (Fig. 6C). In the other cell subsets of NK cells, LPS induced the increased number of Khdc1a-positive, Klra1-positive and Mki67-positive NK cells, while treatment with either Dex or TMP only slightly inhibited the effects of LPS in MKI67-positive NK cells (Fig. S3). These results demonstrated the regulatory effects of TMP on the immune cell subsets in LPS-induced ALI.

Figure 5 Differential remodeling of myeloid cells in LPS-induced acute lung injury after TMP treatment.

(A and B) Sub-clustering of myeloid cells based on marker genes (A). Heatmap showing the top three genes for each myeloid cell sub-cluster (B). (C) LPS-increased Vcan+ monocytes in the C3 sub-cluster were significantly repressed by Dex treatment, while TMP had a mild inhibitory effect.

Figure 6 The effects of TMP treatment on the cell subsets of NK cells.

(A and B) Sub-clustering of NK cells based on marker genes (A). A heatmap showing the top three genes for each NK cell sub-cluster (B). (C) LPS increased Siva+ NK cells in the C5 sub-cluster, which were significantly attenuated by either Dex or TMP treatment.

Identification of molecular changes caused by TMP in epithelial cell subsets

Epithelial cells are considered as the main cell type damaged in ALI (Long, Mallampalli & Horowitz, 2022; Peters et al., 2014). Therefore, we investigated which epithelial cell subsets were affected by TMP in its protective effects against cell damage (Wang et al., 2021; Sauler et al., 2022). As shown in Fig. 7A, the epithelial cells were divided into eight sub-clusters based on marker genes. The top three genes for each cell clusters of epithelial cells are shown in a heatmap (Fig. 7B). We found that these cell subsets were generally present in different samples and groups, suggesting clustering of epithelial cells based on cell types from different mice (Figs. S5A and S5B). LPS significantly decreased the number of Krt14 high basal cells, which is a subset of cells within the C1 sub-cluster, and this decrease was reversed by TMP treatment. However, no significant differences were observed between the LPS and LPS+Dex groups (Fig. 7C). In other cell subsets, no significant cytotoxic effects were observed in the LPS group, except for the mixed lineage cell subset. Treatment with Dex led to further reduction in cell numbers, and no significant protective effects were found in the mixed lineage cell subset following the treatment with TMP when compared with the LPS group (Fig. S4). These results indicated that the treatment with TMP protected against LPS-induced cell damage in Krt14 high basal epithelial cell subset.

Figure 7 Identification of molecular changes in epithelial cell subsets caused by TMP.

(A and B) Sub-clustering of epithelial cells based on marker genes (A). A heatmap showing the top three genes for each epithelial cell sub-cluster (B). (C) LPS-induced decrease in Krt14 high basal cells in the C1 sub-cluster was significantly repressed by TMP treatment, but not by treatment with Dex. (D) Analysis of critical transcription factors with increased activities in Krt14 high basal cells using scenic analysis. CKrt14 high basal refers to a cluster of Krt14 high basal cells. (E) TMP treatment attenuated the LPS-induced reduction of FOSL1 expression in basal cells.

To examine the molecular changes in basal cells after TMP treatment, we also conducted differential gene expression analysis and performed GO, KEGG and GSEA analyses by using the differentially expressed genes to derive meaningful biological insights from our large-scale genomic dataset. GO analysis revealed that the down-regulated genes were enriched in DNA repair, intrinsic apoptotic signaling pathways in response to DNA damage and intrinsic apoptotic signaling pathways. The up-regulated genes were enriched in mitotic spindle organization, mitotic cell cycle checkpoint and cell cycle phase transition. This indicated that the treatment with TMP might regulate the cellular physiological processes related to cell survival and cell cycle progression in basal epithelial cells (Fig. S5C). KEGG analysis showed that the down-regulated genes were enriched in purine metabolism and the P53 signaling pathway, while the up-regulated genes were enriched in the ErbB, TGF-β and MAPK signaling pathways in basal cells (Fig. S5D). GSEA analysis indicated that the FOXO, ErbB and TGF-beta signaling pathways were positively associated with TMP treatment in basal cells (Fig. S5E). These results showed that the ErbB and TGF-β signaling pathways are likely to play important roles in mediating the TMP-protective effect against cell damage induced by LPS in basal epithelial cells. To identify the critical transcription factors affected by TMP in epithelial cells, we performed scenic analysis to determine the changes in their activities, especially in Krt14 high basal cells. We found that the activity of FOSL1, an essential transcription factor involved in regulating several cellular physiological processes, was increased significantly in Krt14 high basal cells (Fig. 7D). In addition, TMP treatment attenuated the LPS-induced decrease in FOSL1 expression in Krt14 high basal cells (Fig. 7E). These results suggest that FOSL1 is likely to act as a crucial downstream target in TMP protection of basal epithelial cells against LPS-induced cell death.

Identification of molecular changes caused by TMP in endothelial cell subsets

Endothelial cell dysfunction is also a key pathological change in ALI (Matthay et al., 2019; He et al., 2021). Therefore, we examined which subsets of endothelial cells could be affected by TMP (Geldhof et al., 2022; Lou et al., 2022). The endothelial cells were divided into six sub-clusters (Fig. 8A). A heatmap of the top three genes for each cell clusters of endothelial cells are shown (Fig. 8B). We found that these six sub-clusters were generally present in different samples and groups (Figs. S7A and S7B). As shown in Fig. 8C, LPS treatment decreased the number of LECs, which was a subset of cells within the C6 sub-cluster. TMP treatment significantly repressed the effects of LPS and increased the percentage of LECs, while treatment with Dex led to a decreased number of these cells. In the other cell subsets, LPS also significantly decreased the cell percentage of Atypical Capillary (aCap) and arterial endothelial cells, whereas no significant protective effects were found after treatment with TMP (Fig. S6). These results indicated that TMP could protect the LECs from damage, while Dex treatment may have negative effects on them.

Figure 8 Identification of molecular changes in endothelial cell subsets after treatment with TMP.

(A and B) Sub-clustering of endothelial cells based on marker genes (A). A heatmap showing the top three genes for each endothelial cell sub-cluster (B). (C) LPS decreased the number of lymphatic endothelial cells (LECs), which was significantly mitigated by TMP treatment, while Dex treatment led to decreased numbers of LECs. (D) Analysis of critical transcription factors with increased activities in LECs using scenic analysis. Clymphatic ECs refers to a cluster of lymphatic endothelial cells. (E) TMP treatment weakened the inhibitory effects of LPS on JunB expression in LECs, but not after treatment with Dex.

Differential gene expression analysis was performed on the LECs, and GO, KEGG and GSEA analyses were conducted by using the differentially expressed genes. GO analysis showed that the down-regulated genes were enriched in the proteasomal protein catabolic process and Ras protein signal transduction, while the up-regulated genes were enriched in mitochondrial organization and response to oxidative stress in the LECs. This implied that TMP might affect the mitochondrial functions and the status of oxidative stress (Fig. S7C). KEGG analysis revealed that the down-regulated genes were enriched in protein processing in the endoplasmic reticulum and ubiquitin-mediated proteolysis, while the up-regulated genes were enriched in DNA replication, proteasome and RNA degradation (Fig. S7D). GSEA analysis indicated that biosynthesis of the amino acids as well as cysteine and methionine metabolism were positively associated with TMP treatment in the LECs (Fig. S7E). These results indicated that TMP treatment might influence protein synthesis and modification as well as amino acid metabolism. Scenic analysis revealed that JunB, a transcription factor involved in regulating oxidative stress, was activated significantly in LECs (Fig. 8D). TMP treatment significantly increased the expression of JunB in LECs, while Dex treatment did not have the same effect (Fig. 8E). These results suggested that TMP was likely to provide better protection for the survival of the endothelium in LPS-induced ALI.

Discussion

ALI is a serious lung condition characterized by excessive inflammation and alveolar epithelial cell necrosis, which can ultimately lead to acute respiratory distress syndrome and pulmonary fibrosis (Matuschak & Lechner, 2010). Dex can partially alleviate this condition and is a commonly used drug in clinical practice, but it can have serious side-effects that limit its widespread use (Noreen, Maqbool & Madni, 2021). TMP, a regulator of oxidative stress, has been shown to reduce the expression of pro-inflammatory factors and alleviate lung injury (Lin et al., 2022). However, the specific mechanisms by which TMP affects the progression of lung injury are largely unknown and require further investigation. In the present study, we used scRNA sequencing analysis to explore the potential protective mechanism of TMP against LPS-induced ALI. Our results imply that TMP exerted its effects on immune, epithelial and endothelial cells, ultimately leading to the attenuation of lung injury.

The present study focused on the immunomodulatory effects of TMP on immune cells. Previous studies have highlighted the roles of immune inflammatory cells in the progression of ALI (Villaseñor-Altamirano et al., 2023; Shi & Pamer, 2011; Franklin, Connolly & Hussell, 2022). CD8+ T cells have been implicated in the development of lung injury through the release of cytotoxic mediators and induction of apoptosis in lung epithelial cells (Villaseñor-Altamirano et al., 2023). Similarly, monocytes and NK cells have been shown to contribute to lung inflammation and injury through the release of pro-inflammatory cytokines and cytotoxic molecules (Shi & Pamer, 2011; Franklin, Connolly & Hussell, 2022). Our findings support these previous studies, emphasizing the importance of these immune cell subsets in the pathogenesis of ALI. We found that TMP suppressed the activation of CD8+ naive T cells, Vcan-positive monocytes and Siva-positive NK cells, while increasing the number of follicular helper T cells. Siva, a regulatory protein that plays a role in cell apoptosis, was initially identified by its interaction with CD27 and is known as CD27 binding protein. It is reported that the BCL2L1 subtype is expressed in human neutrophils and Siva can regulate neutrophil apoptosis through BCL2L1 after being stimulated by inflammatory mediators (Walmsley et al., 2011). Siva can also inhibit p53 activity by stabilizing the interaction between MDM2 and p53 and promoting the degradation of p19ARF, thereby enhancing cell proliferation (Du et al., 2009). Vcan (proteoglycan core protein) also known as chondroitin sulfate proteoglycan or CSPG5, can bind to hyaluronic acid and participate in the synthesis of GAGs as well as the expression and regulation of key sulfation enzymes (Venkatesan et al., 2014). Excessive Vcan expression has been associated with increased cell proliferation and anti-apoptotic ability in fibroblasts. p53 acts as the main downstream target of Vcan-regulated cell apoptosis (LaPierre et al., 2007). These findings suggest that TMP may exert its suppressive effects on the inflammatory process by modulating these immune cell populations in LPS-induced ALI.

Another important finding in this study was that TMP protected epithelial cells from injury and increased the population of Krt14 high basal cells. These protective effects were mediated by the upregulation of FOSL1 in epithelial cells. Previous studies have highlighted the importance of basal cells in maintaining lung homeostasis and repairing injury (Wu et al., 2022). FOSL1, a member of the activator protein-1 (AP-1) transcription factor superfamily, is known to play a crucial role in regulating various cellular processes such as cell survival, proliferation, oxidative stress and resistance to chemotherapy (Talotta, Casalino & Verde, 2020). It has been reported that FOSL1 participates in mediating the antiapoptotic pathways and contributes to resistance against kinase inhibitors treatment (Khedri et al., 2024). Overexpression of FOSL1 leads to the enhancement of cell proliferation and cell cycle progression by activation of several pathways (Al-Khayyat et al., 2023). However, the effects of FOSL1 on the survival of lung epithelial cells are still unknown. Our results show that FOSL1 acts as a key mediator in TMP ability to protect against LPS-induced death of epithelial cells. These results imply that TMP may promote the survival and integrity of epithelial cells, at least in part, by inducing the expression and activation of FOSL1.

Additionally, the present study demonstrated that TMP could increase the number of LECs and protect them against injury by upregulating JunB expression. In addition to epithelial cells, endothelial cells are also damaged in ALI, and their dysfunction is a crucial pathological change in this condition (Matthay et al., 2019; He et al., 2021). LECs are the main cellular structure that makes up the lymphatic vessel walls. Previous studies have reported that LECs play crucial roles in maintaining fluid homeostasis, regulating lymphocyte recirculation and immune cell trafficking in the lungs (Hu et al., 2024). JunB has been shown to have protective effects against oxidative stress-induced cellular damage (Son et al., 2010). Oxidative stress can potentially disrupt lymphatic function and contribute to tissue edema and inflammation (Mukohda, Mizuno & Ozaki, 2020). In our study, we found that LPS significantly reduced the number of LECs, which was reversed by TMP treatment. In contrast, treatment with Dex led to a further reduction in LECs when compared to the LPS group. Our results were consistent with previous reports that TMP could protect endothelial cell from damage. JunB proto-oncogene, a crucial member of the dimeric AP-1 complex, is known as an immediately early gene that is activated in response to a wide variety of cellular stimuli and it is essential in regulating various physiological, including cell proliferation, apoptosis, senescence and metastasis (Ren et al., 2023). The roles of JunB in different cells are not always consistent. It has been reported that JunB positively regulates the proliferation of embryonic fibroblast cells by promoting S to G2/M transition through cyclin A activation (Andrecht et al., 2002). However, it negatively regulates cell proliferation by inhibition of G1-S transition in HeLa cells through inhibition of the cyclin D1 promoter (Bakiri et al., 2000). In human lymphoma cells, JunB has been shown to protect against oxidative stress-induced cell death (Son et al., 2010). Consistently, our results showed that JunB was inhibited by administration of LPS in endothelial cells, which is increased by TMP treatment. However, it was decreased by Dex treatment. These results suggest that the upregulation of JunB by TMP in LECs can enhance their antioxidant capacity and preserve their function, thereby contributing to the resolution of lung injury. However, further experimental studies are required to validate the findings from the data analysis and to confirm these conclusions under different experimental conditions.

Conclusions

In conclusion, this study provides new insights into the mechanisms underlying the protective effects of TMP in ALI. TMP demonstrated multifaceted protective effects against lung injury by its modulation of the immune response, promotion of epithelial cell survival and enhancement of endothelial cell antioxidant capacity. These findings support the potential therapeutic use of TMP in the management of ALI and lay the foundation for further investigations into its molecular mechanisms and its potential clinical applications.

Supplemental Information

Supplemental Information 1 Code.

Supplemental Information 2 Supplemental Figures.

Supplemental Information 3 Author Checklist.

We are grateful to Dr. Dev Sooranna, Imperial College London, for English language edits of the manuscript.

Abbreviations

ALI Acute lung injury

TMP 2,3,5,6-tetramethylpyrazine

Dex Dexamethasone

LPS Lipopolysaccharide

scRNA-seq Single cell RNA sequencing

TNF-α Tumor necrosis factor alpha

IL-1β Interleukin-1 beta

IL-6 Interleukin-6

BALF Bronchoalveolar lavage fluid

GO Gene Ontology

KEGG Kyoto Encyclopedia of Genes and Genomes

FOSL1 FOS Like 1

JunB JunB Proto-Oncogene

NK Natural killer

DNT Double negative T cells

Teff Effector T cells

Treg Regulatory T cells

Inflam-Mφ Inflammatory macrophages

RTM-Mφ Resident tissue macrophages

GSEA Gene Set Enrichment Analysis

LECs Lymphatic endothelial cells

aCap Atypical capillary

EMT Epithelial-mesenchymal transition

Additional Information and Declarations

Competing Interests

Author Contributions

Animal Ethics

DNA Deposition

Data Availability

The authors declare that they have no competing interests.

Mingyan Wu performed the experiments, analyzed the data, prepared figures and/or tables, authored or reviewed drafts of the article, and approved the final draft.

Shanmei Wang performed the experiments, authored or reviewed drafts of the article, and approved the final draft.

Xiaolan Chen analyzed the data, prepared figures and/or tables, and approved the final draft.

Li Shen analyzed the data, authored or reviewed drafts of the article, and approved the final draft.

Jurong Ding analyzed the data, authored or reviewed drafts of the article, and approved the final draft.

Hongbin Jiang conceived and designed the experiments, prepared figures and/or tables, authored or reviewed drafts of the article, and approved the final draft.

The following information was supplied relating to ethical approvals (i.e., approving body and any reference numbers):

Tongji University guidelines and the Animal Welfare Act Regulations (Ethics approval number: Q20-269Y).

The following information was supplied regarding the deposition of DNA sequences:

The sequencing is available at GEO: GSE276682.

The following information was supplied regarding data availability:

The code is available in the Supplemental File.

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
