# Peer review of "Single-cell transcriptome analysis reveals cellular reprogramming and changes of immune cell subsets following tetramethylpyrazine treatment in LPS-induced acute lung injury"

_PeerJ, doi:10.7717/peerj.18772_

## Round 0.1 · original submission · Major Revisions

The authors are requested to carefully revise the manuscript and answer the questions raised by the reviewers.

Reviewer 1 ·

Basic reporting

This study investigated TMP effects on lung inflammation and cell death induced by LPS. In this study, they attempted to verify the efficacy of TMP by confirming changes in immune cells and lung cells using an ALI animal model.
But, it must be improved in professional English and full sentences. For example, in 123 to 134, the author did not use full sentences.

Experimental design

To assert their opinion, they must provide more solid data and experimental results through a multi-faceted approach. However, this article seems to lack experimental evidence to support their claims.

The authors insist that Dex or TMP treatment attenuated LPS-induced acute lung injury, edema, and inflammatory cell infiltration based on only one histological examination. To verify this, they should present more detailed and critical data, such as immune cell staining and highlighting the edema with an indication of the site.

In this article, there are several errors or missing information on abbreviated terms. For example, what are ARDS (Figure 1G) and Clymphatic ECs (Figure 5E)?

In Figure 2, which sample was used for A? If you used a control sample, why are the results different between Figures 2A and 2E?

You only presented CD8+ T cells and follicular helper T cells in Figure 3. Why did you handle Treg and other T helper cells such as Th1, Th2 , and Th17. Please present other T cell populations because T cell numbers decreased in all conditions compared with the control.

Based on your result, TMP increased Tfh number. what is happening in the B cells population, Plasma cells, and Immunoglobulins (expression and subclass).

In the GO, KEGG, and GSEA analyses, why do you focus on some genes? Also, the explanations are not sufficient.

Validity of the findings

They wanted to demonstrate the efficacy of TMP, but there seems to be a lack of experiments and claims to support this. To address this, they need to revise their strategy, provide more solid data to assert the efficacy of TMP, and include sufficient explanations and definitions of abbreviations to make the data understandable. Additionally, it is necessary to provide adequate explanations and reference materials regarding the use of terms.

Additional comments

It could be useful if some supportive data and detailed information were included. However, some data should be included in their figures or supplementary materials. Considering this article aims to demonstrate the screening of TMP via single-cell transcriptome analysis, it should have more references based on their data. Additionally, to strongly support their points, more critical data and a multi-faceted approach are necessary.

Additionally, the introduction should be improved by including more details on the immune cells, lung cells, and genes addressed in this article. Furthermore, the English should be thoroughly checked for any grammatical or syntax errors.

Reviewer 2 ·

Basic reporting

In this paper, the authors investigate cellular reprogramming induced by Tetramethylpyrazine (TMP) in a mouse model of LPS-induced acute lung damage.

Experimental design

They utilized single-cell RNA sequencing to examine the dynamic changes in the lung tissue microenvironment, focusing on immune cells, epithelial cells, and endothelial cells after TMP treatment. Notably, the study identifies distinct subclusters within these cell types and finds that TMP treatment increases the number of follicular helper T cells, basal epithelial cells, and lymphatic endothelial cells, while decreasing CD8+ naive T cells, Vcan-positive monocytes, and Siva-positive NK cells

Validity of the findings

The research is well-structured, and the data analysis is generally robust, providing valuable insights into how TMP mitigates LPS-induced lung damage. However, there are areas for improvement.

Additional comments

1. The study found that neither dexamethasone nor Tetramethylpyrazine significantly protected against cell loss in LPS-induced acute lung injury. Could you clarify the reasoning behind this finding?

2. Figure 1 does not include the statistical analysis for panel B, which makes it difficult for readers to interpret the results.

3. Some images, such as Figure 3C and 4E, have insufficient resolution, leading to a lack of clarity. These figures need to be improved for clarity.

4. The manuscript contains a few grammatical errors that should be corrected during revision, such as on page 11, line 4.

Reviewer 3 ·

Basic reporting

This article effectively explores cellular reprogramming following Tetramethylpyrazine treatment in LPS-induced acute lung injury. The writing is precise and presents ideas clearly, facilitating the reader’s understanding of complex concepts. The citations provide ample background and context, highlighting the significance of the findings within the broader field. The article is well-organized and easy to follow. However, the availability of raw data remains unclear.

Experimental design

The researchers conducted single-cell RNA sequencing on a mouse model of acute lung injury induced by Lipopolysaccharide. It compared the effects of treatments with TMP and Dex on lung tissue, focusing on changes in immune cells, epithelial cells, and endothelial cells. The research also analyzed Tetramethylpyrazine's effects on cell subtypes, gene expression patterns, and potential transcription factor regulation.

Validity of the findings

This study provides original research on the protective mechanisms of Tetramethylpyrazine against Lipopolysaccharide-induced injury using single-cell RNA sequencing. It identifies the immune, epithelial, and endothelial cell subtypes regulated by TMP and explores potential transcription factors involved. This comprehensive approach offers valuable and novel insights into this serious disease, significantly enhancing the article's contribution to the field. However, the study primarily focuses on data analysis and would benefit from additional experimental validation to support its findings.

Additional comments

While the study is reasonably well-organized, several issues should be addressed to enhance the article's impact:
1. The authors should clearly specify where the raw data is accessible,facilitating its use and citation by researchers in related fields.
2. The primary focus of the study is to demonstrate TMP's protective effect on lung injury; however, the rationale for including the Dex group needs to be clearly explained.
3. The SCENIC analysis indicates changes in several transcription factors following TMP treatment. The authors should explain their reasoning for selecting Fosl1 and Junb as potential downstream targets.

---

## Round 0.2 · Minor Revisions

The authors are requested to carefully revise the manuscript and answer the questions raised by Reviewer 1.

The UMAP data presented in Fig. 1A appears unusual. The authors should carefully review and validate this part of the analysis.

Reviewer 1 ·

Basic reporting

The authors effectively demonstrated the effects of TMP through their experiments. In particular, their presentation of changes in immune cell subclasses provides valuable insights into the anti-inflammatory effects of TMP, making this study a significant contribution for future research in this area.

Experimental design

The information regarding the concentration of LPS used in the animal experiments appears to be somewhat unclear. Additionally, it is unclear at what specific time points Dex and TMP were administered following the initial LPS treatment. The authors should also specify the treatment volumes for LPS, Dex, and TMP.

Validity of the findings

The study successfully showed changes in specific immune cell populations following TMP treatment. This provides an important foundation for understanding the mechanisms by which TMP modulates immune responses and highlights the potential for further investigation into its immunoregulatory effects.

Additional comments

Overall, the study offers meaningful insights into how TMP treatment affects immune cell changes and inflammation induced by LPS. By examining the alterations across various immune cell populations, the authors shed light on the mechanisms of TMP’s anti-inflammatory effects. However, in Fig. 1A, the immune cell proportion in lung tissue appears unusually high. I would recommend the authors verify and explain this observation.

Reviewer 2 ·

Basic reporting

No comments.

Experimental design

No comments.

Validity of the findings

No comments.

Additional comments

The authors have responded to me appropriately and had the manuscript professionally edited. I have no additional comments.

Reviewer 3 ·

Basic reporting

The authors implemented my recommendation by making their raw data publicly available.

Experimental design

No comment

Validity of the findings

The authors conducted additional experiments and addressed the study's limitations. The revised paper is now acceptable.

Additional comments

The authors addressed my questions and revised the manuscript accordingly. No further comments are required.

---

## Round 0.3 · accepted · Accept

After revisions, two reviewers agreed to publish the manuscript. There was one reviewer left with a minor revision, and I think the author has responded adequately. I also reviewed the manuscript and found no obvious risks to publication. Therefore, I also approved the publication of this manuscript.